# Network-based visualisation of frequent sequences

**László Bántay**⬚*◎, **János Abonyi**◎

HUN-REN-PE Complex Systems Monitoring Research Group, University of Pannonia, Veszprém, Hungary

◎ All these authors are contributed equally to this work.
* bantay.laszlo@mk.uni-pannon.hu

## Abstract

Frequent sequence pattern mining is an excellent tool to discover patterns in event chains. In complex systems, events from parallel processes are present, often without proper labelling. To identify the groups of events related to the subprocess, frequent sequential pattern mining can be applied. Since most algorithms provide too many frequent sequences that make it difficult to interpret the results, it is necessary to post-process the resulting frequent patterns. The available visualisation techniques do not allow easy access to multiple properties that support a faster and better understanding of the event scenarios. To answer this issue, our work proposes an intuitive and interactive solution to support this task, introducing three novel network-based sequence visualisation methods that can reduce the time of information processing from a cognitive perspective. The proposed visualisation methods offer a more information rich and easily understandable interpretation of sequential pattern mining results compared to the usual text-like outcome of pattern mining algorithms. The first uses the confidence values of the transitions to create a weighted network, while the second enriches the adjacency matrix based on the confidence values with similarities of the transitive nodes. The enriched matrix enables a similarity-based Multidimensional Scaling (MDS) projection of the sequences. The third method uses similarity measurement based on the overlap of the occurrences of the supporting events of the sequences. The applicability of the method is presented in an industrial alarm management problem and in the analysis of clickstreams of a website. The method was fully implemented in Python environment. The results show that the proposed methods are highly applicable for the interactive processing of frequent sequences, supporting the exploration of the inner mechanisms of complex systems.

## Introduction

Frequent sequence pattern mining is widely used to extract knowledge from event log files, e.g., it has been successfully applied for user activity pattern analysis [1], workload prediction [2], malware behaviour analysis [3], and prediction of transition probability [4]. Frequent sequence pattern mining algorithms do not provide sufficient information on the relationship and context of the large number of sequences extracted, making the interpretation and utilisation of the results difficult. In the case of association rules, there are three main types of post-

**Data Availability Statement:** The used data and the python source files are available publicly in the following GitHub repository: https://github.com/laszlobantayPE/NbVFS.

**Funding:** This work has been implemented by the OTKA 143482 (Monitoring Complex Systems by

goal-oriented clustering algorithms) and the TKP2021-NVA-10 projects with the support provided by the Ministry of Culture and Innovation of Hungary from the National Research, Development and Innovation Fund, financed under the 2021 Thematic Excellence Programme funding scheme. The funders had no role in study design, data collection and analysis, decision to publish, or preparation of the manuscript.

**Competing interests:** The authors have declared that no competing interests exist.

processing methods for the results to solve this kind of problem, pruning, grouping, and visualisation [5]. These post-processing methods can be applied to sequences as well.

Visualisation can be a powerful tool, especially when developing machine learning models [6], for example, pruning algorithms [7]. A network-like representation allows one to see and focus on relevant information without reading the data in detail [8]. A well-designed visualisation method can also allow the grouping of data by a relevant attribute. For example, sequential patterns contain the order of events, along with their frequency, which supports the development of event-propagation models. A wide range of sequential pattern visualisation techniques have been developed in the past and were summarised and compared in a study [9]. These techniques were individual representations [10], flow diagrams [11], aggregated pattern visualisations [12], placement strategies [13], and episode visualisations [14]. The comparison showed that a sequence pattern visualisation technique provides valuable information (support, confidence, sequence relations, etc.) or is easy to read and understand. With an increasing number of patterns, readability quickly worsens, affecting lucidity. If the goal is to analyse a system with many events containing many relevant sequences, this problem must be solved.

Based on the available metrics of the visualisation methods [15], we have chosen three key attributes that can serve as a metric to measure the conformity of the applicability. These are *Information content*, *Readability*, and *Flexibility*.

- **Information content**: More content gives a better and more complete view of the system, but above a certain level, it causes indistinctness. Information can be the support values, the confidence values of sequences and their transitions, the relationship between connected and nonconnected sequences, like the parent-child relationship, similarity, or the group where the sequence belongs.

- **Readability**: The balance between information content and readability is a critical issue in data visualisation concepts, especially in the case of a large amount of data to process.

- **Flexibility**: From a data set, different knowledge extraction directions can be performed. Scalability and the opportunity to tailor the method to the actual task are essential. A good visualisation concept has filtering options to perform goal-oriented analysis tasks.

The motivation of our work is to suggest a solution that provides a good balance with respect to all attributes. It is also worth noting that most solutions are not open source and custom modifications are not easy. Although there are solutions that successfully overcome these shortcomings [16], we believe that our proposed method is a useful addition to these techniques. From a clarity point of view, finding the balance between information content and readability, and from an applicability point of view, allowing good flexibility in tailoring goal-oriented analysis tasks.

The use of network theory in the case of frequent sequence pattern post-processing supports the identification of relevant event types, such as unifying and polarising events [17]. Existing visualisation techniques focus on the interpretation of well-defined processes, for example, in [18] and in [19]. The main difference between our approach and existing ones is that the proposed method does not aim to visualise one process model but rather to identify the different and common elements of parallel process models. In industrial log files, where technology is complex, stored events do not unequivocally describe existing subprocesses. One solution is to create sublog files based on frequently co-occurring events [20]. Our technique supports the selection of events that describe a particular process, enabling the definition of targeted process models.

This work proposes three visualisation methods that allow quick access to the context of the data without eliminating any relevant information—for example, relatively rare, but important

**Table 1. Comparison of visualization techniques.**

| Method | Information content | Readability | Flexibility |
|---|---|---|---|
| Individual representation [10] | + | / | + |
| Flow diagram [11] | + | + | - |
| Aggregated pattern [12] | + | + | ++ |
| Placement strategy [13] | + | + | ++ |
| Episode visualization [14] | / | / | ++ |
| NBVFS | + | + | ++ |

events—during postprocessing. These methods are NBVFS-WG (Network-Based Visualisation of Frequent Sequences—Weighted Graph), NBVFS-CM (Network-Based Visualisation of Frequent Sequences—Confidence-Based Multidimensional scaling), and NBVFS-TM (Network-Based Visualisation of Frequent Sequences—Transaction-Based Multidimensional scaling).

The proposed method was developed in a Python environment, ensuring focused analysis through tailorability. Furthermore, specific solutions can be built on the basic principle of the method. The key idea is to interpret the connection of frequent sequences as a network, using sequence metrics as network attributes and as inputs for similarity measurement. This approach can be considered a network embedded with side information [21] and can be a powerful analysis tool that pairs with the temporal skeletonization of sequential data discussed in [22]. The characteristics of the networks provided represent the characteristics of the frequent sequences obtained from the data set. For example, the width of an edge can be proportional to the confidence value of the transition between the nodes that directly represent subsequent sequences. The proposed method performs well with respect to the three requirements mentioned above (Table 1):

The application of MDS ensures that as much information as possible is kept; dimension reduction can be applied succesfully in solving high-dimensional problems, for example, to interpret large amount of association rules [23]. The network-like representation provides good readability. Finally, the open-source character gives excellent flexibility.

The contributions of this work are the following:

- the detailed process description of methods NBVFS-WG, NBVFS-CM, and NBVFS-TM is introduced, along with the theoretical background;

- the gained network-based visualizations of the frequent sequences are demonstrated and discussed;

- a summary of the proposed methods is provided with suggestions for future research directions.

The remainder of the paper is organised as follows. First, the theoretical background and a detailed description of the proposed method are presented. Second, use cases are provided with two different data sources, along with a discussion of the results. In the end, the proposed method is summarised and future research directions are identified.

## The proposed algorithms for the network-based visualisation of frequent sequences

This section discusses the proposed visualisation algorithms with a related theoretical background.

The method aims to provide an informative visualisation of the frequent patterns of sequences. The concept of the proposed visualisation process can be seen in Fig 1. The source is a data set where items are ordered and grouped in sequences in the form of transactions. The transactional database is processed with a sequence pattern mining algorithm, and the output is the set of frequent sequences. The frequent sequences are completed with several attributes, like confidence or parent sequence. These attributes form the basis of the visualisation process. Three ways were chosen to visualise the networks: confidence-based (NBVFS-WG and NBVFS-CM) and transaction-based (NBVFS-TM).

- **NBVFS-WN** (**N**etwork-**B**ased **V**isualization of **F**requent **S**equences—**W**eighted **N**etwork): this method uses a confidence-based adjacency matrix and the calculated metrics of the sequences. It results in a weighted network where those sequences are connected that are direct extensions of each other. The weight is the confidence value of the transition between the two connected sequences. This kind of visualisation helps to understand the conditions and consequences of occurring events.

- **NBVFS-CM** (**N**etwork-**B**ased **V**isualization of **F**requent **S**equences—**C**onfidence-based **M**DS projection): this method uses a similarity calculation, using transition confidence values for similarity measurement. The adjacency matrix must be enriched regarding the non-directly connected sequences using transitive distance calculation. The positions of the nodes on the network are calculated with Multidimensional scaling, and the more similar the two sequences are, the closer they will be presented on the network. This kind of network representation contains more information about the relationship of sequences.

- **NBVFS-TM** (**N**etwork-**B**ased **V**isualization of **F**requent **S**equences—**T**ransaction-based **M**DS projection): the positions of the nodes are calculated with MDS identically to the NBVFS-CM method. The process is more or less the same, but the similarity calculation is based on the overlap of their common supporting transactions. This approach is closer to process mining, as transactions can be taken as traces and can provide feedback to the mining process.

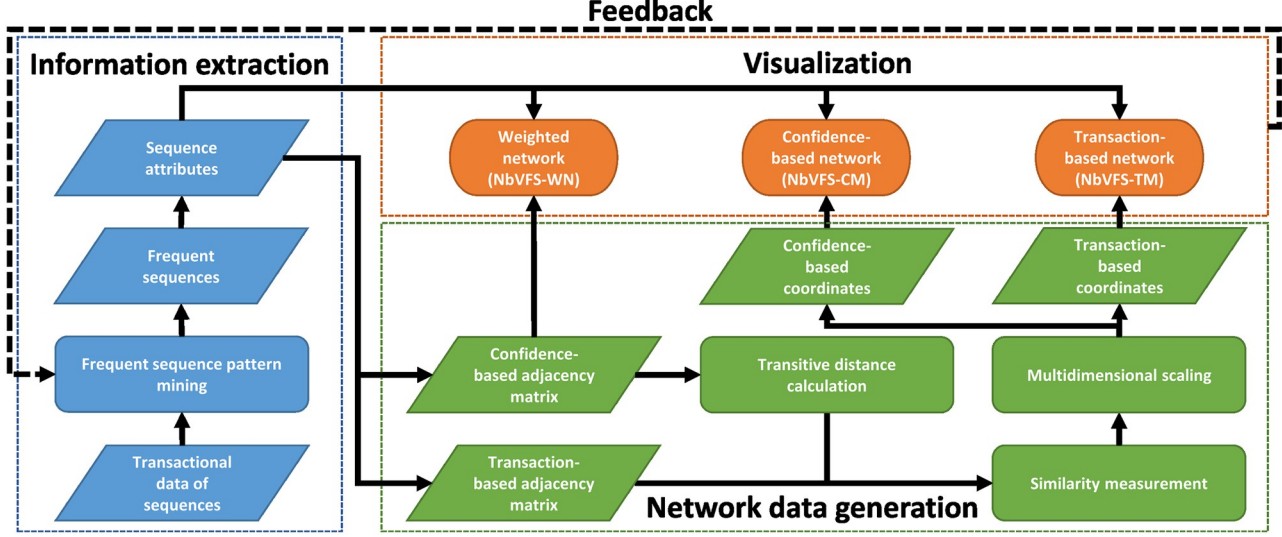

**Fig 1. The concept of the visualization process.** After the frequent sequences are produced from the data set, their attributes are calculated. These attributes are the inputs of the three visualization algorithms, NBVFS-WG, NBVFS-CM, and NBVFS-TM.

## Frequent sequence pattern mining

A sequence is an ordered list of itemsets [24], in this work, the items are $e$ events. Let us denote the set of all events with $I$, $I = \{e_1, \ldots, e_k, \ldots, e_p\}$. The set of $T = \{T_1, \ldots, T_i, \ldots, T_n\}$ transactions contain events, $T_i$ is the $i$-th transaction, $T_i = \{e_1, \ldots, e_p\}$ and $T_i \subseteq I$. A sequence $\phi_i = \{e_1, \ldots, e_k\}$ is a sequence in $T_i$, $\phi_i \subseteq T_i$. The set of all sequences is $\phi = \{\phi_1, \ldots, \phi_i, \ldots, \phi_l\}$. There are two important attributes of sequences, support and confidence. The support of a sequence is the number of times it occurs or the number of transactions containing it. Let $S^{\phi_i}$ be the set of transactions that support $\phi_i$. The cardinality of $S^{\phi_i}$ is the support of $\phi_i$:

$$| S^{\phi_i} | = Sup(\phi_i). \tag{1}$$

One sequence can occur in many transactions, but one transaction can occur only once. To consider a sequence as frequent, we must define a threshold *minSup*:

$$\{\phi_i \in \phi_{freq} : Sup(\phi_i) > minSup\}, \tag{2}$$

where $\phi_{freq}$ is the set of all frequent sequences and $\phi_{freq} \subseteq \phi$.

The confidence (*Conf*) is the conditional probability of the occurrence of a sequence of $k$ length, assuming that its predecessor of $k - 1$ length (or parent sequence) has already occurred. If $\phi_j = (\phi_i \rightarrow e_j)$, so $\phi_j$ is a one event extension of $\phi_i$, and $S^{\phi_j}$ is the set of transactions that support $\phi_j$, considering Eq 1 we can state, that

$$Conf(\phi_i \rightarrow \phi_j) = \frac{Sup(\phi_j)}{Sup(\phi_i)} = \frac{| S^{\phi_j} |}{| S^{\phi_i} |}. \tag{3}$$

From the obtained frequent sequences, information has to be extracted as much as possible. Frequent sequential pattern mining algorithms need some input parameters [25]. These parameters are:

- *minsup*: the minimum support value threshold to consider a sequence as frequent.

- *max gap*: Adjusting the gap to M means that an M-1 event is allowed between the consecutive items of the pattern.

- *min pattern length*: the minimum number of events that the found patterns should contain.

- *max pattern length*: the maximum number of events the found patterns should contain.

A data set must be prepared to contain all frequent sequences and all the necessary information describing the relationship between them. The goal is to visualise the connection network of the sequences by connecting the parent-child sequences, such as $\phi_i$ and $\phi_j$, by using the attributes of the sequences as network parameters. In this work, two visualisation methods were applied, a confidence-based and a similarity-based one.

Consider the sequences $\phi_i$ and $\phi_j$ as nodes, where $\phi_j$ as the extension of one event of $\phi_i$, $\phi_j = (\phi_i \rightarrow e_j)$. $G(V, E)$ is a weighted directed network, where $V(G)$ is the set of nodes, $E(G)$ is the set of edges [26]. If $(\phi_i, \phi_j) \in E(G)$ (a directed edge from node $\phi_i$ to $\phi_j$ exists), the weight of the edge $(\phi_i, \phi_j)$ is considered equal to the confidence value of the transition $(\phi_i \rightarrow \phi_j)$, and $N$ is the number of nodes, then the $\phi_i\phi_j$ related value of the adjacency matrix $A = (a_{i,j})_{N \times N}$ is

$$a_{i,j} = Conf(\phi_i \rightarrow \phi_j). \tag{4}$$

All sequences are collected around their joint starting event in this weighted network. There are visualisation options to emphasise the attributes of the network. For example, the confidence of edges and nodes can be indicated by their width or the sequence length with

colour, gaining a visually informative network. The limitation of this network is that it does not handle the connection between the different subnetworks. Its only organising principle is the direct connections between events, starting from their shared first event; the network is undirected.

The method NBVFS-CM discovers the relationship between non-directly connected sequences to create a directed network by calculating their similarity values. As $S^{\phi_i}$ and $S^{\phi_j}$ are sets, the Jaccard similarity measurement can be used.

## Network visualisation of sequences

The principle of the NBVFS-WN method is to connect the sequences with their direct extensions, weighted by the confidence value of the transition, resulting in a similarity value:

$$sim(\phi_i, \phi_j) = \frac{\mid S^{\phi_i} \cap S^{\phi_j} \mid}{\mid S^{\phi_i} \cup S^{\phi_j} \mid}. \tag{5}$$

Considering that $\phi_j$ is the one event extension of $\phi_i$, we can state that $S^{\phi_j} \subseteq S^{\phi_i}$, so $\mid S^{\phi_i} \cap S^{\phi_j} \mid = \mid S^{\phi_j} \mid$ and $\mid S^{\phi_i} \cup S^{\phi_j} \mid = \mid S^{\phi_i} \mid$. This means that the similarity between two sequences is the ratio of the cardinality of their supporting transactions. That is, according to Eq 3, the confidence of transition ($\phi_i \rightarrow \phi_j$):

$$sim(\phi_i, \phi_j) = Conf(\phi_i \rightarrow \phi_j). \tag{6}$$

The adjacency matrix of the weighted network can be used to gain the similarity values directly. Still, it has to be enriched with values between nondirectly connected sequences.

It can be supposed that the 2-length and 4-length subsequences of a 5-length sequence are related. Transitive distance calculation is a proper tool for obtaining missing values. The transitive distance method is based on the similarity of nodes [27]. This method helps calculate the similarity between nodes not connected directly (non-neighbours). A sequence can be taken as the connected nodes of a network (in a directed way), where the sequences are the nodes. Consider the transition system $\phi_i \rightarrow \phi_j \rightarrow \phi_k$, where $\phi_j$ is the one-event extension of $\phi_i$ and $\phi_k$ is the one-event extension of $\phi_j$. Although $\phi_i$ and $\phi_k$ are not connected directly, they are in relation through $\phi_j$. Using Eq 6 and the theory of transitive distance calculation,

$$sim(\phi_i, \phi_k) = sim(\phi_i, \phi_j) \times sim(\phi_j, \phi_k) = Conf(\phi_i \rightarrow \phi_j) \times Conf(\phi_j \rightarrow \phi_k). \tag{7}$$

This equation shows that the confidence value of the sequence transitions can be used to calculate the transitive distance-based similarity between nondirectly connected nodes. With this method, the visualisation of the network is enriched with nondirect connections, using multidimensional scaling (MDS) to define the relative position of the sequences on the network. The main principle of this approach is similar to hyperbolic embedding of networks [28], but uses different similarity measures.

In this work, *Metric (or classical)* MDS was used. MDS is a tool to visualise items in a high-dimensional feature space by mapping them to a low-dimensional data space (mainly 2D), based on the similarity of the items in the original space [29]. It discovers the hidden structure of the items by preserving similarity information, which is the pairwise distance between the items. Like other dimension reduction-based visualisation methods, MDS tries to minimise a

stress function $E$, that is, using the square error cost.

$$E_{metricMDS} = \frac{1}{\sum_{i<j}^{N} d_{i,j}^{*2}} \sum_{i<j}^{N} (d_{i,j}^* - d_{i,j})^2.$$

(8)

In the proposed method, $d_{i,j} = 1 - sim(\phi_i, \phi_j)$ (the original distance between $\phi_i$ and $\phi_j$), and $d_{i,j}^* = \| \mathbf{y}_i - \mathbf{y}_j \|$, where $\mathbf{y}_i$ and $\mathbf{y}_j$ are the coordinates of sequences $\phi_i$ and $\phi_j$ in the visualised network. Adding the $\mathbf{Y} = [\mathbf{y}_1, \ldots, \mathbf{y}_i, \ldots, \mathbf{y}_N]$ vector of the sequence coordinates gained to the sequence attributes already collected, the network can be plotted. The visualisation options are similar to those of the weighted network. The support of the sequences can be represented by the size of the nodes, while the confidence of the succeeding sequence with the width of the edge connected to it.

The similarity of the two sequences can also be calculated based on the number of shared supporting transactions (NBVFS-TM). In this method, unlike NBVFS-CM, the similarity values cannot be obtained directly. Information about related supporting transactions is needed, which is represented by an N × n matrix, where N = $|\phi_{freq}|$ and n $= | S^{\phi_{freq}} |$, and $S^{\phi_{freq}}$ is the set of transactions that support frequent sequences. We get an adjacency matrix N × N filled with the number of common supporting transactions of the sequences. The Jaccard similarity can be calculated from this matrix, and the MDS-driven plotting of the network is identical to that of NBVFS-CM.

The three visualisation methods can be formalised as an algorithm now that the necessary theoretical background is discussed.

**Algorithm 1** NBVFS

```
Input: T                         ▷ T is the set of transactions
{φ_freq}, S^φ ← FSPM(T,M)         ▷ M:{minsup,max gap,min pattern
length,max pattern length}
i ← 1
while i ≤ N do                    ▷ N =|φ_freq|
  j ← 1
  if method is Confidence-based then            ▷ NBVFS-WN or
NBVFS-CM
    while j ≤ N do
      A ← Conf(φ_i → φ_j)                ▷ A = (a_{i,j}) ∈ ℝ^{N×N} is the adja-
cency matrix of φ_freq
      if method is NBVFS-WN then
        V(G), E(G), w ← A              ▷ nodes (V), edges (E) and
weights (w) of the network
      else if method is NBVFS-CM then
        A* ←TransDist(A)               ▷ Calculation of transitive
distances
        D = 1-A*                ▷ D is the dissimilarity matrix
        Y(G) ← MDS(D)              ▷ Y is the coordinate vector of
nodes
        V(G), E(G) ← A             ▷ nodes (V) and edges (E) of the
network
      end if
      j ← j + 1
    end while
  else if method is Transaction-based then              ▷ NBVFS-TM
    while j ≤ n do                 ▷ n=| S^{φ_freq} |
```

```
        A ← |S^{φ_i} ∩ S^{φ_j}| / |S^{φ_i} ∪ S^{φ_j}|    ▷ A = (a_{i,j}) ∈ ℝ^{N×n} is the adjacency matrix
of φ_freq and S^{φ_freq}
        D = 1−A                         ▷ D is the dissimilarity matrix
        Y(G) ← MDS(D)                       ▷ Y is the coordinate vector of
nodes
        V(G), E(G) ← A                       ▷ nodes (V) and edges (E) of the
network
        j ← j + 1
    end while
  end if
  i ← i + 1
  end while
  plot G                      ▷ NBVFS-WN: Weighted network, NBVFS-CM/TM:
MDS projection
Output: G                     ▷ Network-Based Visualisation of Frequent
Sequences
```

From a time complexity point of view, as the method assumes that frequent sequences have already been obtained, the computational complexity of the generation of the weighted network and the MDS projections must be discussed separately. If the number of events is $|I|$, then, in the case of the weighted network, the maximal number of edges between the $|I|$ nodes is $|I| \times (|I| - 1)$, that is, the complexity. The complexity of the MDS-based representation depends mainly on the complexity of the MDS algorithm (which is equivalent to the complexity of the distance matrix), that is, $|I|^3$. The complexity of frequent sequence pattern mining is $2^{|I|}$, so in the case of large amounts of events (where the method starts to be really helpful), the exponential characteristic of the sequence pattern mining becomes dominant, so the visualisation part does not cause significant complexity increase compared to the sequence pattern mining itself.

## Results and discussion

The applicability of the methods was validated using two types of data sets. The first is a click-data set of a website that is an ideal input to demonstrate the benefits of NBVFS. The second is an alarm management log file, as this development work was motivated by alarm management. Alarm management is the set of techniques and tools that support the safe and efficient control of industrial processes [30]. In alarm management, the prediction of the probability of event propagation based on historical data is crucial. Analysing event chains that have already occurred allows action scenarios for events that occur in the future. Frequent sequence pattern mining is a widely used tool in alarm management, for example, in alarm suppression methods [31], or in grouping different alarm floods based on historical data using similarity measurement [32]. However, pattern mining alone is insufficient for deep exploration of the actual system from an event-dependence point of view. This work aims to extend the existing toolkit for event data analysis based on the proposed methods and theories.

The weighted network has many available layout options, for example, a "circle" type (the starting event is located in the centre, and the sequences form concentric circles with increasing length outward), or a "hierarchical" type (the starting event is located on top with increasing sequence length downward). The width of the edges represents the confidence of the transition of the two connected sequences. The colours represent the length of a sequence. The size of the node is proportional to the confidence value of the sequence.

The transitive distance-based network provides more information about the relationship of the sequences than a weighted network. The additional information is the relationship between those sequences that are not connected.

**Table 2. Statistics of the source data.**

| Data source | Click data | Industrial alarm data |
|---|---|---|
| Number of events | 1636 | 28536 |
| Number of event types | 14 | 357 |
| Number of traces | 343 | 3956 |
| Average trace length | 5 | 7 |

The developed method also allows filtering on the networks. An option available in both network types (weighted and MDS projection) is to filter on the first event of the connected sequences (one or more), enabling a targeted visualisation. Another option is to draw only those edges above a minimum confidence level, allowing the identification of the most probable event scenarios. The latter is available for MDS projections.

The first source was click data from UWV (Employee Insurance Agency), a Dutch autonomous administrative authority, downloaded from the IEEE—Task Force For Process Mining website [33]. There are the following 14 identified click actions with their representing codes in the networks: Your last employer—0, The dismissal—1, Hours worked—2, Other work/income—3, Your employment history—4, Send data—5, Your personal details—6, Other information—7, Supplement—8, Your situation—9, Your income—10, Your possessions—11, The labour market—12, Your personal information—13. The source file contained data for eight months. However, to save computational time, only part of the data was used to demonstrate the operation of the methods. Based the input and the purpose of the analysis, several algorithms can be used. Our selection was CM-SPAM [34], as this can provide the IDs of the supporting traces, the optional hyperparameters (for example, *minsup*) give good flexibility and a customisation option for the pattern mining task and also provide good computational speed. The input attributes for the algorithm were the following: *minsup* 1%, *max gap* 1, *min pattern length* 1, *max pattern length* 10. Sequence pattern mining resulted in 242 frequent sequences, with lengths between 1 and 8.

The other data source was an alarm management log file from a Hydrofluoric Acid Alkylation (HFA) plant. To meet the terms and conditions of the data source, all events were coded to numbers (which is also required to run the frequent pattern searching algorithms). The plant consists of four production units and more than 400 tags (source of the signals). The distributed control system is a Honeywell product. The file contained more than 200,000 events over four months, that was filtered and reduced to save computational time. The type of events in this work was limited to two, *alarm* and *operator action*. The algorithm used was again CM-SPAM. The input attributes for the algorithm were the following: *minsup* 1%, *max gap* 2, *min pattern length* 1, *max pattern length* 10. As the algorithm needs numbers as input, the events were coded to integers with the following rules: the numbers between 100 and 499 represent *alarm*s, above 1000 *operator action*s. Pattern mining resulted in 420 frequent sequences with a length between one and six. As the number of nodes is significantly high, two frequent sequence-starting events were chosen for an in-depth analysis of the results. The operator actions '1084' and '1085' control the bottom inlet and the steam inlet of the stripper, respectively. Each step of the method was performed in the *Python* programming language. A summary of the data used can be found in Table 2.

## NBVFS-WN

The option of filtered on the start events enables a goal-oriented analysis. This kind of visualisation helps identify frequent trigger events and has quick access to the probability of

occurrence of events that depend on previous events simultaneously. The weighted network has some visible characteristics that are worth discussing. Many edges get thicker heading to the outside of the network, meaning the confidence of the transitions increases with every sequence extension. Events occurring later in the sequence depend highly on preceding events; the network represents a higher-order Markov chain. If this phenomenon occurs in a highly branched sequence line, it indicates the most probable sequence of events, enabling targeted filtering on the network.

**Weighted network created from click-data.** The weighted network (Fig 2) was filtered for starting action *2*, namely "Hours worked". The hierarchical layout was chosen to get maximum readability from a relatively low number of nodes present in the network. The network clearly shows the most probable following actions after "Hours worked", which are "Other work/income" (3) and "Your employment history" (4), with "Supplement" (8) as the most frequent final action.

This network allows to identify action rules very quickly, for example: if actions 2, 3 and 4 occur, the most probable successive action will be 8.

**Weighted network created from alarm management data.** The network produced can be seen in Fig 3. The most probable action to occur in addition to the operator action 1084 is 1085. The majority of sequences represented by network nodes contain these two actions. Interestingly, the most probable action scenarios also have these two actions as end events.

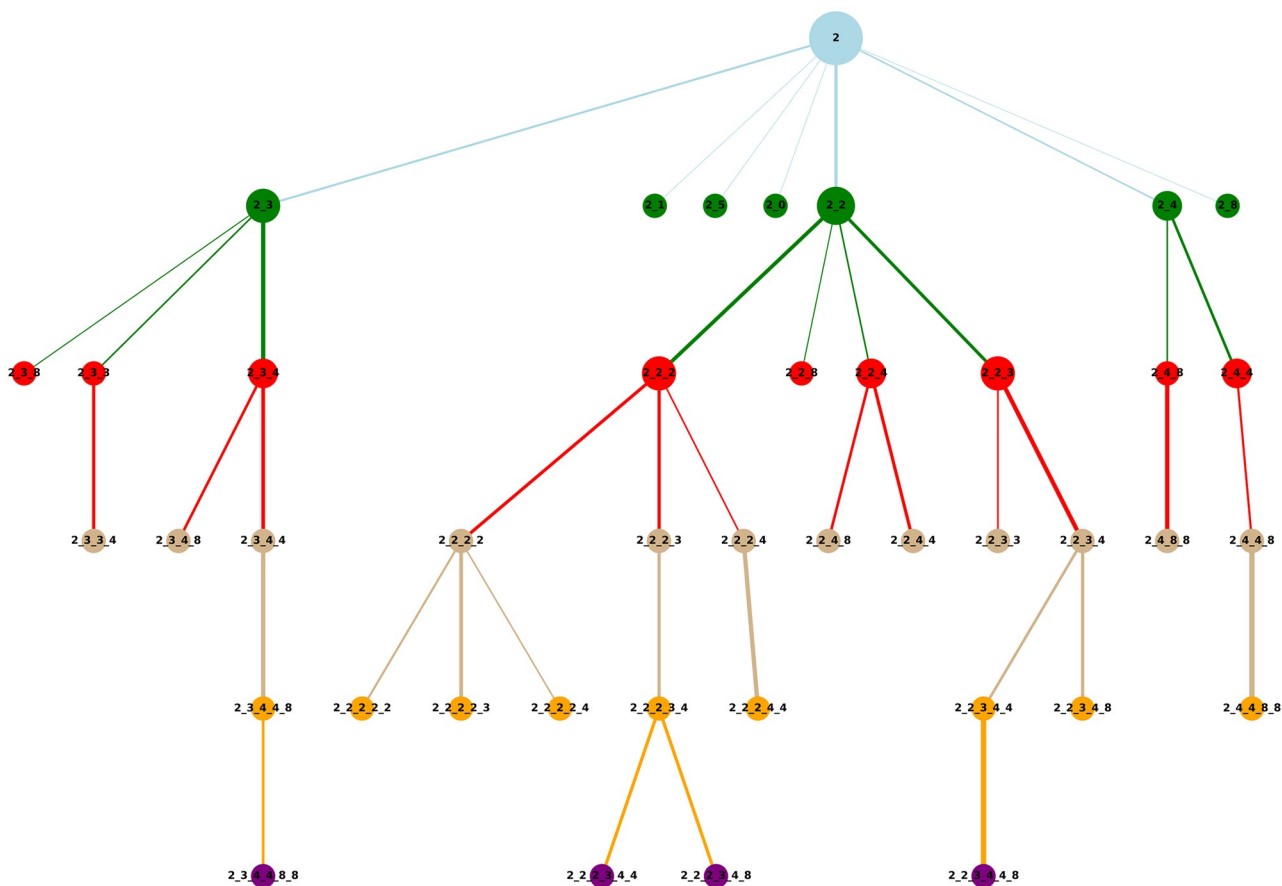

**Fig 2.** Weighted network produced from click-data, filtered on starting event "Hours worked" (2). It is followed mainly by "Other work/income" (3) and "Your employment history" (4). If 2, 3 and 4 four occurs, the most frequent end event is "Supplement" (8).

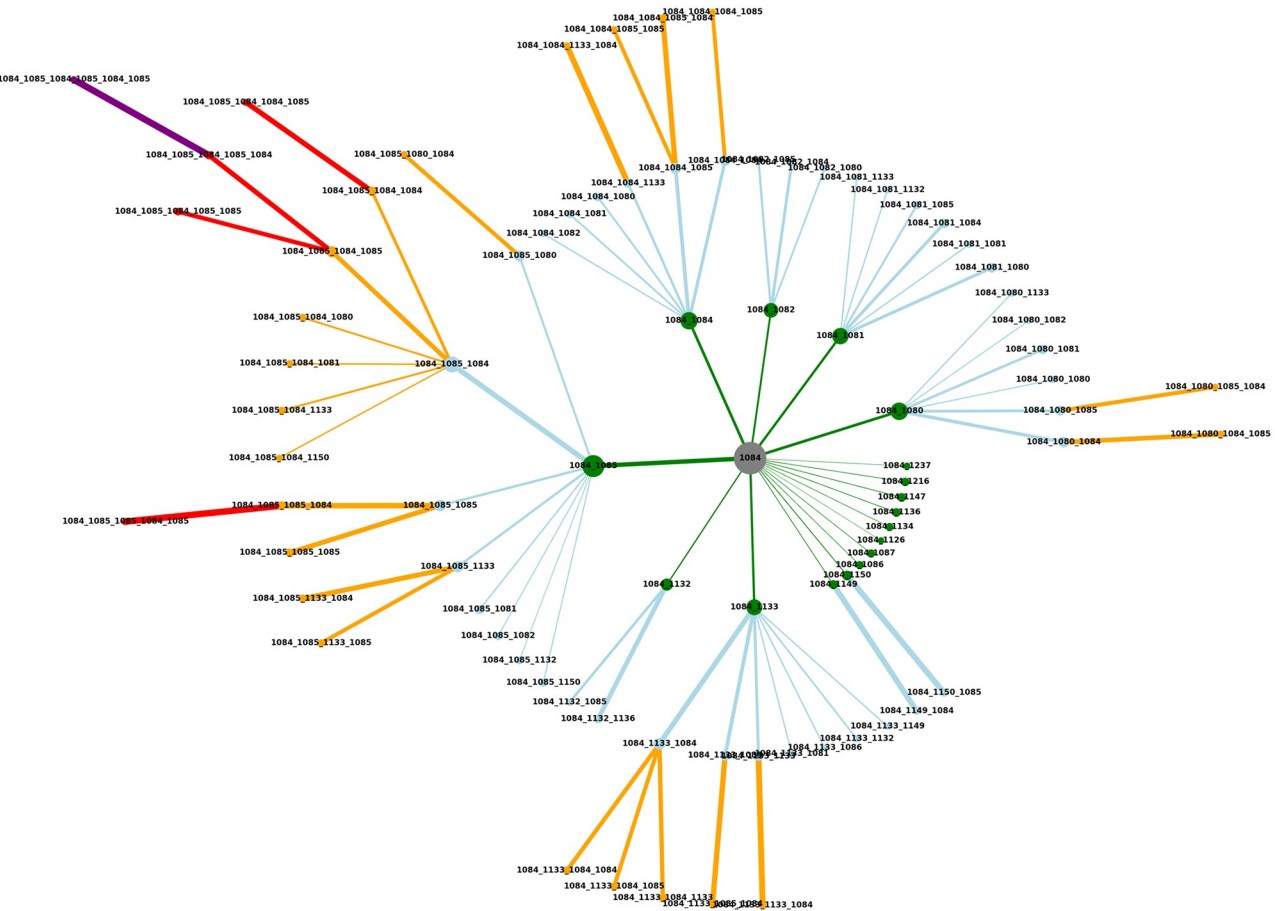

**Fig 3. The weighted network produced from alarm log data, filtered on starting event operator action 1084.** A clear view of the event propagation probability distribution is provided. It gives an informative overview of operators' intervention strategies.

With action 1084 occurring, an extended action sequence can be expected, and this can be clearly seen by checking the sequence routes containing transitions with high probability. By adding the actual state of the system variables when operator actions are triggered, this network can support the development of an early warning system.

## NBVFS-CM and NBVFS-TM

The results of these two methods will be discussed in one section, since although they are both MDS projections based on similarity values, there are some differences between them.

In the NBVFS-CM generated network, the width of the edges represents the confidence of the subsequent node. This confidence value is decreasing and is heading to the periphery of the network. In the case of the transitive distance-based MDS projection(NBVFS-TM), where transition confidence values are used as similarity values, the distance between two nodes represents the confidence of that transition.

Some statements can be made for both types of visualisation. The longer an edge, the lower the confidence of that transition. In addition, the more outgoing edges a parent point has, the longer the edges. It means that the children's confidence is lower if a parent has more children, which is logical. If an event triggers more probable chain events, the probability of those chains

will be reversely proportional to the number of chains. It is valid for points in the middle of the network, where events that occur in many transactions are present. Closer to the network's periphery, the points located close to each other are related by similarity. An attractive area is the "single" points. Those events are not connected, this means they are not a part of any frequent sequence, at least with the actual gap setup, but they occur with the connected ones close to them. In alarm management, these single points can be chattering alarms. If we raise the gap value, the number of isolated points and subnets will be lower; however, new single points will occur.

**MDS projections of the click-data.** Based on Fig 2, the MDS projections were filtered for starting events 2, 3, and 4. There are apparent differences between the two networks (Figs 4 and 5). The nodes in the red boxes in Fig 4 are far from each other, which means that their similarity value is low, at least using confidence values and transitive distance calculation. These nodes in Fig 5 have almost identical coordinates (practically 2_3_3_4 and 3_3_4 have

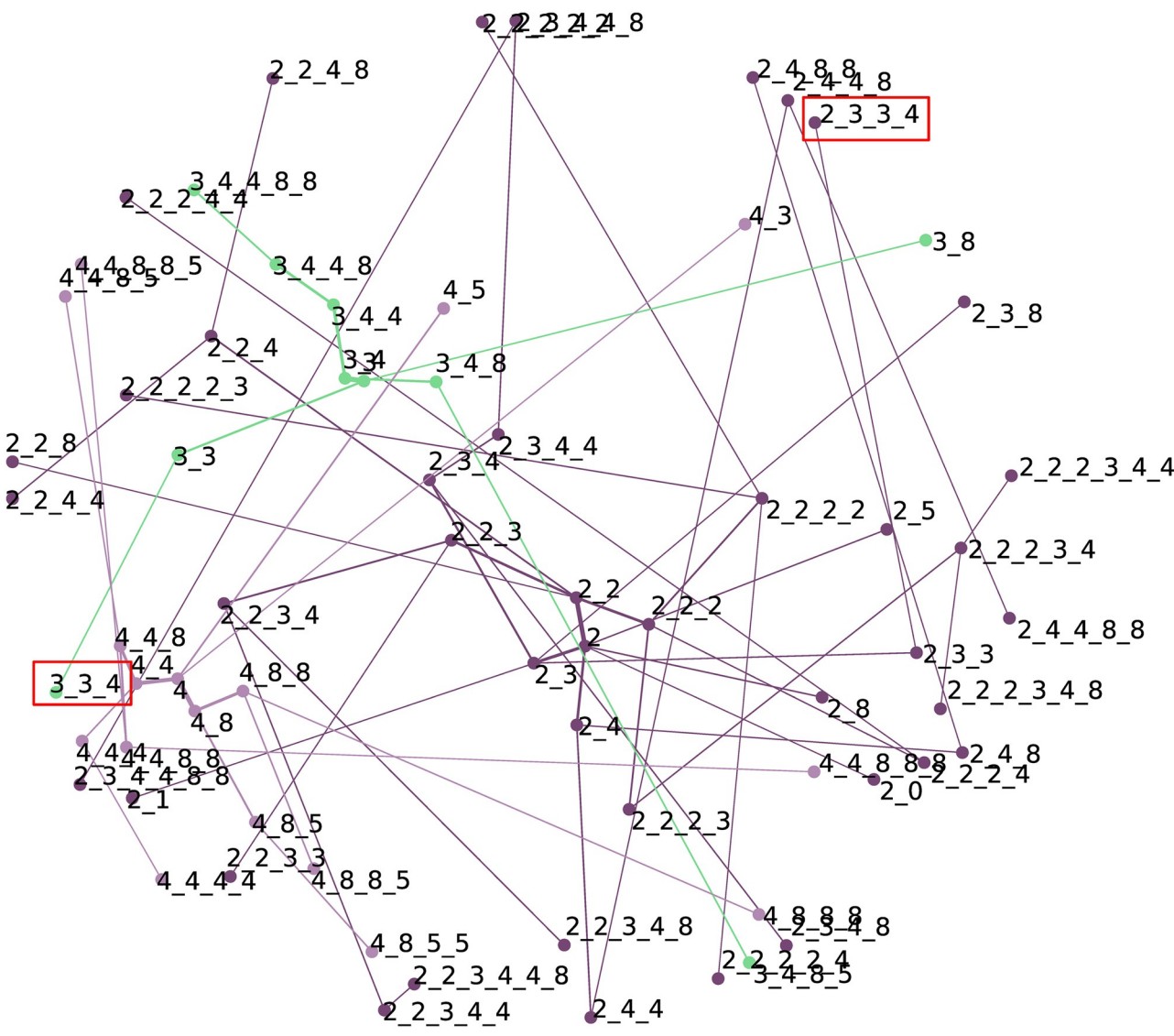

**Fig 4. The confidence-based MDS projection of the click-data.**

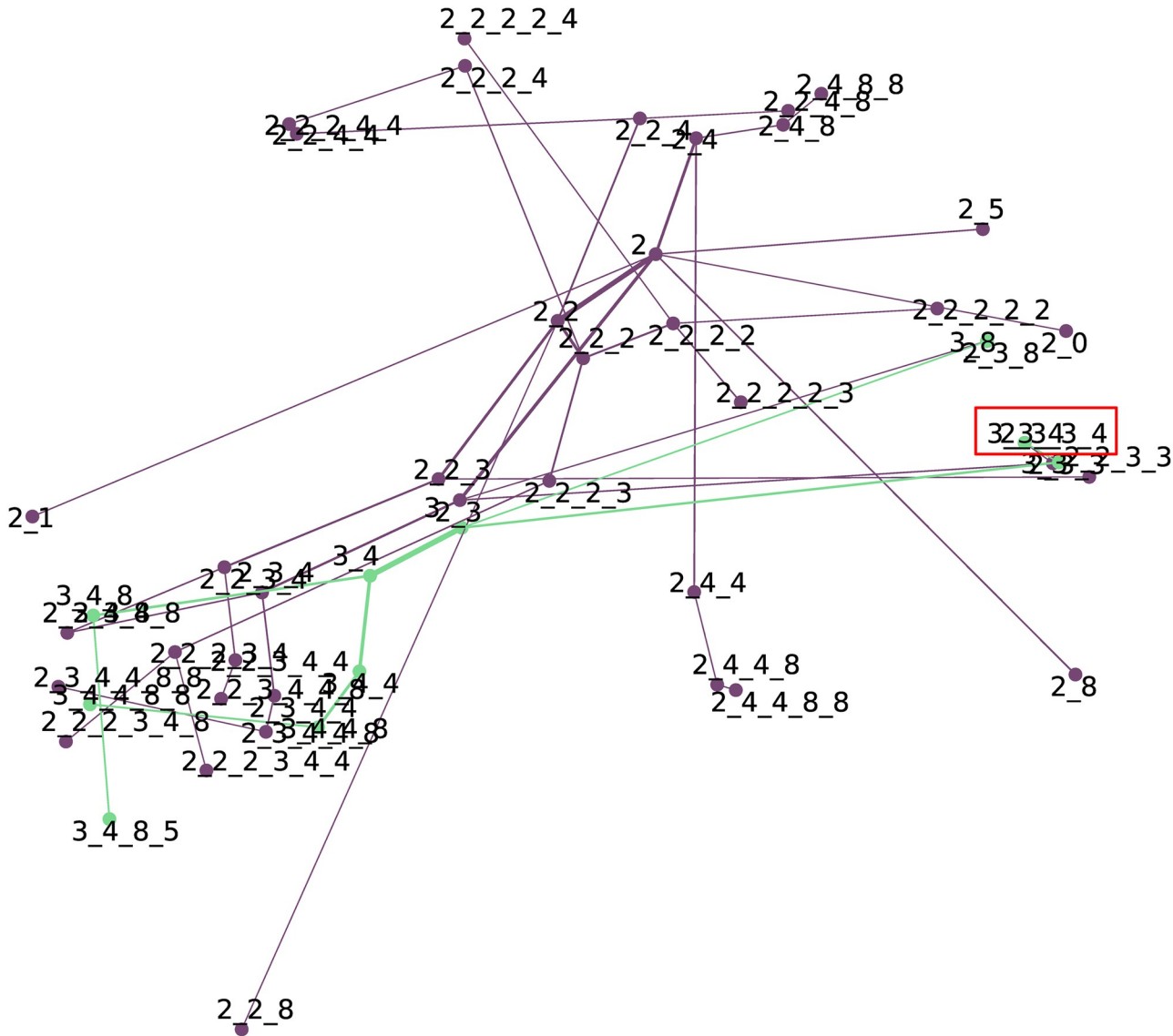

**Fig 5. The transaction-based MDS projection of the click-data.**

the same), which means that they occur most of the time in the same transactions, which is also underlined by Fig 2.

**MDS projections of alarm management data.** There is a clear difference between confidence-based networks (Fig 6) and transaction-based networks (Fig 7). The projection provided by the NBVFS-TM method has denser parts where those sequences are located, which are permutations of the same several events. This approach is closer to Process Mining. In the case of industrial log files, where the transactions are traces and these traces are generated based on timestamps, we can state that points close to each other are occurring close in time. Based on this theory, the length of the nodes represents the average time elapsed between two consecutive sequences. In this way, the traces are represented by the points located close to each other; they can be called event clusters. We can identify multi-cluster triggering events connected

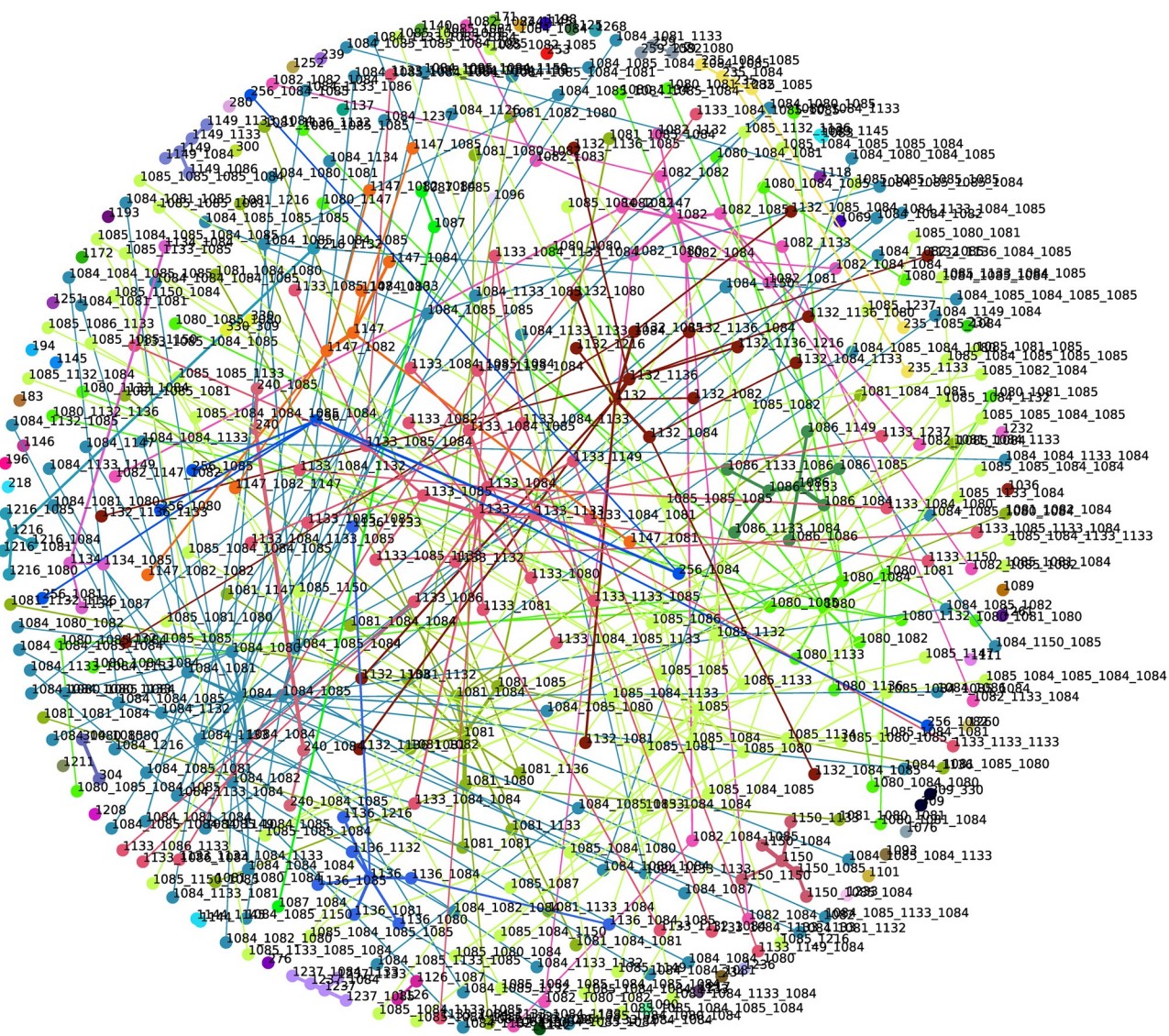

**Fig 6. The network of sequences provided by method NBVFS-CM.** It provides an enriched information visualization compared to the weighted network.

with clearly separated event groups, which can be helpful information in targeted event analysis and process mining tasks.

## Conclusion

The analysis of frequent sequences is an essential technique for the extraction of knowledge from data-based event systems. There are many available methods to explore the connection between sequences of events, and one section is devoted to the visualisation of them. Every visualisation technique has advantages and disadvantages, and the critical factor is finding the balance between information content and readability. Besides these two essential requirements, flexibility is also important and an excellent visualisation-based tool is applicable for diverse analysis tasks. This work aimed to extend this toolkit by applying a network-based

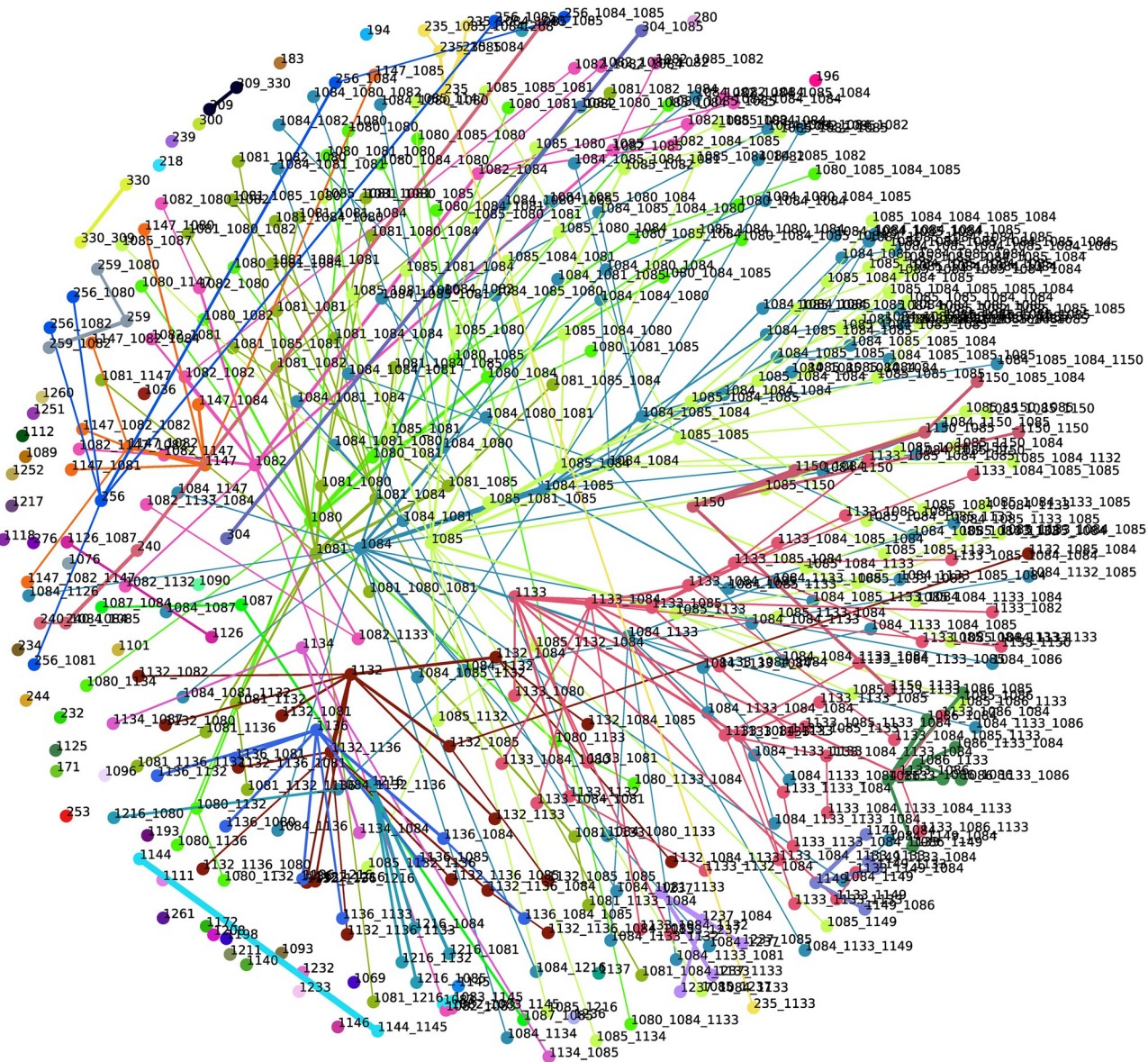

**Fig 7. The network of sequences provided by method NBVFS-TM.** There are clear groups separated from each other, and the sequences within occur in the same transactions.

approach to frequent sequence pattern mining, seeking to fulfil the mentioned requirements as well as possible.

The confidence values of the frequent sequences can be considered similarity values in the case of direct successor sequences. The adjacency matrix can be enriched with transitive distance calculation, gaining similarity values of nondirectly connected sequences. The basis of similarity can be the overlap of the supporting transactions of the sequences, and this approach provides a different view of the relationship of the sequences. A network can be created using Multidimensional scaling, where the distances between the nodes represent their similarity.

The NBVFS substeps are similar to three main groups of association rule (AR) visualisation techniques, which are one step away from the sequences. The end products of the proposed

method, the networks, stand for the network-based view of AR, proving that NBVFS offers a wide range of built-in visualisation options (this work focused on the network-based ones). The three types of networks provide different types of knowledge extraction. The weighted network identifies frequent trigger events by representing support and confidence values and the length of sequences on the same network. The network provided by NBVFS-CM is an extension of the weighted network, adding the similarity of sequences as a new dimension. The resulting network of the NBVFS-TM method is based on a slightly different approach. Using the overlap of supporting transactions is one step closer to process mining. The visible groups of sequences on the network represent traces. This method can be applied to validate trace generation rules or help define them if the event database is not well structured and labelled. Last, but not least, as an open-source solution, other types of targeted visualisation are also possible. The source code will be shared on GitHub to make it available to the broad scientific community.

The potential in similarity measurement is that it can be combined with other measures. This network-like visualisation of time-series-type event databases has many possible applications. The introduced method is a goal-oriented analysis tool to extract useful knowledge from the event database by visualising the event sequences. Due to its interactive character, it can successfully support iterative data analysis tasks. For example, it enables the adjustment of the frequency sequence pattern mining parameters. The proposed visualisation method allows for quick recognition of relevant event chains and key events with their most important attributes, such as support and confidence. This kind of information interpretation besides the support in cognitive information processing, can speed up the parameter identification step of machine learning model building, for example. It can also validate trace rules and other parameters used in process discovery tasks with respect to process mining. It has to be customised for the exact purpose of the task.

Although the methods are already valuable additions to sequence visualisation techniques, there are clear directions for future research. One option is the segmentation of the network. It would give a more targeted interpretation of the event relationships using network attributes like size or density. Combining similarity measurements and their aggregation with other measures is also promising.

## Author Contributions

**Conceptualization:** János Abonyi.

**Methodology:** László Bántay, János Abonyi.

**Supervision:** János Abonyi.

**Visualization:** László Bántay.

**Writing – original draft:** László Bántay.

**Writing – review & editing:** László Bántay, János Abonyi.

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
