## [Decision Letter · Decision Letter 0]

21 Jul 2023

PONE-D-23-17033Network-based visualization of frequent sequencesPLOS ONE

Dear Dr. Bántay,

Thank you for submitting your manuscript to PLOS ONE. After careful consideration, we feel that it has merit but does not fully meet PLOS ONE’s publication criteria as it currently stands. Therefore, we invite you to submit a revised version of the manuscript that addresses the points raised during the review process.

We look forward to receiving your revised manuscript.

Kind regards,

Praveen Kumar Donta, Ph.D.

Academic Editor

PLOS ONE

Journal Requirements:

2. In your Methods section, please include additional information about your dataset and ensure that you have included a statement specifying whether the collection and analysis method complied with the terms and conditions for the source of the data.

4. Please note that funding information should not appear in any section or other areas of your manuscript. We will only publish funding information present in the Funding Statement section of the online submission form. Please remove any funding-related text from the manuscript.

Reviewers' comments:

Reviewer's Responses to Questions

**Comments to the Author**

1. Is the manuscript technically sound, and do the data support the conclusions?

Reviewer #1: Partly

Reviewer #2: Yes

2. Has the statistical analysis been performed appropriately and rigorously? 

Reviewer #1: No

Reviewer #2: Yes

3. Have the authors made all data underlying the findings in their manuscript fully available?

Reviewer #1: Yes

Reviewer #2: Yes

4. Is the manuscript presented in an intelligible fashion and written in standard English?

Reviewer #1: No

Reviewer #2: Yes

5. Review Comments to the Author

Reviewer #1: 1. Please underscore the scientific value added to your paper in your abstract and introduction.

2. Some more recent schemes are also available; how do authors justify that the methods employed are appropriate?

3. The most recent papers need to be addressed in the related work sections.

4. what is the time complexity of proposed algorithms?

5. The motivation of the proposed method should be stated in the introduction.

6. The method proposed by the authors should be compared with the recent existing methods and explained to highlight the

advantages of the new method.

8. How much performance improved in terms of percentage? Please use the abstract to explain this precisely.

9. The language needs to be polished, some grammatical mistakes need to correct.

10. Some formatting errors exist. Correct the entire manuscript with proper formatting.

11. The results are not enough to show the efficiency of the proposed work, so, please extend the results parts to include further experiments.

12. Use high quality images for better visibility.

Reviewer #2: In this manuscript, authors aimed to extend excellent visualization-based by applying a network-based approach to frequent sequence pattern mining, seeking to fulfill the mentioned requirements as well as possible. Authors suggested to address the following comments and suggestions when preparing the revised version:

= Abstract: section needs to be re-drafted to be self-contained means it has to clearly show the hypothesis, methodology, techniques and tools used, and the results obtained.

= Keywords: Authors suggested to add some keywords by selecting relevant terms. Keywords play important role in the appearance of the manuscript in scholars search which will give it more hits and more citations.

= What assumptions authors made during the simulation phase of this research work? If there is any.

= Authors suggested to go through the following references and they MAY make use of them in updating the introduction and the related work sections:

- Ali Akbar Movassagh, Jafar A. Alzubi, Mehdi Gheisari, Mohamadtaghi Rahimi, Senthil kumar Mohan, Aaqif Afzaal Abbasi, Narjes Nabipour, “Artificial neural networks training algorithm integrating invasive weed optimization with diferential evolutionary model” Journal of Ambient Intelligence Humanized Computing, https://doi.org/10.1007/s12652-020-02623-6

- Omar A. Alzubi, Jafar A. Alzubi, Mohammed Alweshah, Issa Qiqieh, Sara Al-Shami, Manikandan Ramachandran, “An Optimal Pruning Algorithm of Classifier Ensembles: Dynamic Programming Approach” Neural Computing & Applications, 2020.

- Jafar A. Alzubi "Diversity Based Improved Bagging Algorithm" In Proceedings of the International Conference on Engineering & MIS 2015 (ICEMIS '15). Istanbul - Turkey.

- Jafar A. Alzubi, "Diversity-Based Boosting Algorithm". International Journal of Advanced Computer Science and Applications, Vol. 7, No. 5, 2016.

- O. A. Alzubi, J. A. Alzubi, S.Tedmori, H. Rashaideh, and O. Almomani "Consensus-Based Combining Method for Classifier Ensembles" The International Arab Journal of Information Technology, Vol. 15, No. 1, 2018.

= Conclusion: The conclusion should be abstracted so authors need to consider re-drafting it.

= Authors need to confirm that all acronyms are defined before being used for first time.

= Authors need to confirm that all mathematical notations are defined when being used for first time.

= Authors suggested to proofread the manuscript after addressing all comments to avoid any typo, grammatical, and lingual mistakes and errors.

= Authors advised to make sure that the format of all references is matching and complying with journal requirements and format.

6. PLOS authors have the option to publish the peer review history of their article (what does this mean?). If published, this will include your full peer review and any attached files.

Reviewer #1: **Yes: **Shashank Singh

Reviewer #2: No

---

## [Author Response · Author response to Decision Letter 0]

6 Nov 2023

Dear Reviewers,

Thank you for your time and efforts to review our article. Based on your valuable comments and suggestions, amendments are made to the manuscript. These amendments surely increase the overall quality of our work. Please find the detailed answers to your findings below.

Reviewer #1: 

1. Please underscore the scientific value added to your paper in your abstract and introduction.

Thank you for this suggestion. Indeed, the aim and added value of our work was not underlined properly. The abstract and introduction is extended now with this aspect.

2. Some more recent schemes are also available; how do authors justify that the methods employed are appropriate?

Thank you for this question. The introduction was extended and redrafted to show the existing techniques of sequence pattern visualisations, and to show the novelty and usefulness of our method by comparing it to existing methods based on three key attributes. 

3. The most recent papers need to be addressed in the related work sections.

Thank you for this note, some recent works were added to the introduction.

4. What is the time complexity of proposed algorithms?

The input of the network are events that are connected. The number of events is |I|. The method assumes that sequence patterns are already gained. Complexity of sequence pattern mining is 2|I|. The network visualisation part can be split in two: the weighted network and the MDS representation. In the case of the weighted network, the maximal number of edges between |I| nodes is |I|*(|I|-1), that is the complexity. The complexity of the MDS-based representation depends mainly on the complexity of the MDS algorithm (which is equivalent to the complexity of the distance matrix), that is |I|3. In the case of large amounts of events (where the method starts to be really helpful), the exponential characteristic of the sequence pattern mining becomes dominant, so the visualisation part does not cause significant complexity increase compared to the sequence pattern mining itself. The discussion of the complexity issue is presented now in the manuscript.

5. The motivation of the proposed method should be stated in the introduction.

Thank you, the motivation is emphasised more now.

6. The method proposed by the authors should be compared with the recent existing methods and explained to highlight the advantages of the new method.

Thank you for the suggestion, the difference between our method and the recent ones is discussed in the introduction now.

8. How much performance improved in terms of percentage? Please use the abstract to explain this precisely.

The method aims to increase the processing time of sequential pattern mining results and to support the selection of relevant related events in the case of subprocess exploration tasks. As it is hard to define a standard value for human cognitive abilities, this question cannot be answered properly. However, if the performance is measured by the information content, we can state that our method allows the visualisation of multiple information, it can significantly lower the time consumption of the post-process activity. We tried to emphasise more the application areas and benefits of the proposed method in the abstract and in the introduction.

9. The language needs to be polished, some grammatical mistakes need to correct.

10. Some formatting errors exist. Correct the entire manuscript with proper formatting.

Thank you for these recommendations, the entire manuscript was checked according to it.

11. The results are not enough to show the efficiency of the proposed work, so, please extend the results parts to include further experiments.

Thank you for this suggestion, we made efforts to emphasise more the benefits and application areas/conditions of the suggested method. We truly believe that the provided experimental results underline the scientific and real life application value of our work. The source code will be shared on GitHub to make the method available for the wide scientific community.

12. Use high quality images for better visibility.

Thank you for this note, Fig2 and Fig3 are reproduced in high quality now.

Reviewer #2: In this manuscript, authors aimed to extend excellent visualization-based by applying a network-based approach to frequent sequence pattern mining, seeking to fulfill the mentioned requirements as well as possible. Authors suggested to address the following comments and suggestions when preparing the revised version:

= Abstract: section needs to be re-drafted to be self-contained means it has to clearly show the hypothesis, methodology, techniques and tools used, and the results obtained.

Thank you for this note, the abstract is extended now to touch these topics.

= Keywords: Authors suggested to add some keywords by selecting relevant terms. Keywords play important role in the appearance of the manuscript in scholars search which will give it more hits and more citations.

We fully agree with this approach, unfortunately the PLOS ONE Latex template does not contain a keywords section.

= What assumptions authors made during the simulation phase of this research work? If there is any.

Thank you for the question. After analysing metrics of visualisation methods, we have chosen three attributes to grade our method: Information content, Readability and Flexibility. This aspect is emphasised more now in the Introduction.

= Authors suggested to go through the following references and they MAY make use of them in updating the introduction and the related work sections:

- Ali Akbar Movassagh, Jafar A. Alzubi, Mehdi Gheisari, Mohamadtaghi Rahimi, Senthil kumar Mohan, Aaqif Afzaal Abbasi, Narjes Nabipour, “Artificial neural networks training algorithm integrating invasive weed optimization with diferential evolutionary model” Journal of Ambient Intelligence Humanized Computing, https://doi.org/10.1007/s12652-020-02623-6

- Omar A. Alzubi, Jafar A. Alzubi, Mohammed Alweshah, Issa Qiqieh, Sara Al-Shami, Manikandan Ramachandran, “An Optimal Pruning Algorithm of Classifier Ensembles: Dynamic Programming Approach” Neural Computing & Applications, 2020.

- Jafar A. Alzubi "Diversity Based Improved Bagging Algorithm" In Proceedings of the International Conference on Engineering & MIS 2015 (ICEMIS '15). Istanbul - Turkey.

- Jafar A. Alzubi, "Diversity-Based Boosting Algorithm". International Journal of Advanced Computer Science and Applications, Vol. 7, No. 5, 2016.

- O. A. Alzubi, J. A. Alzubi, S.Tedmori, H. Rashaideh, and O. Almomani "Consensus-Based Combining Method for Classifier Ensembles" The International Arab Journal of Information Technology, Vol. 15, No. 1, 2018.

Thank you for the suggestions, one article was found useful and was added to the Introduction section.

= Conclusion: The conclusion should be abstracted so authors need to consider re-drafting it.

Thank you for the suggestion, the Conclusion is re-drafted now.

= Authors need to confirm that all acronyms are defined before being used for first time.

= Authors need to confirm that all mathematical notations are defined when being used for first time.

= Authors suggested to proofread the manuscript after addressing all comments to avoid any typo, grammatical, and lingual mistakes and errors.= Authors advised to make sure that the format of all references is matching and complying with journal requirements and format.

Thank you, the manuscript was double checked and amended regarding your suggestions.

---

## [Decision Letter · Decision Letter 1]

13 Dec 2023

PONE-D-23-17033R1Network-based visualization of frequent sequencesPLOS ONE

Dear Dr. Bántay,

Thank you for submitting your manuscript to PLOS ONE. After careful consideration, we feel that it has merit but does not fully meet PLOS ONE’s publication criteria as it currently stands. Therefore, we invite you to submit a revised version of the manuscript that addresses the points raised during the review process.

We look forward to receiving your revised manuscript.

Kind regards,

Praveen Kumar Donta, Ph.D.

Academic Editor

PLOS ONE

Reviewers' comments:

Reviewer's Responses to Questions

**Comments to the Author**

1. If the authors have adequately addressed your comments raised in a previous round of review and you feel that this manuscript is now acceptable for publication, you may indicate that here to bypass the “Comments to the Author” section, enter your conflict of interest statement in the “Confidential to Editor” section, and submit your "Accept" recommendation.

Reviewer #1: All comments have been addressed

Reviewer #2: All comments have been addressed

Reviewer #3: (No Response)

2. Is the manuscript technically sound, and do the data support the conclusions?

Reviewer #1: Partly

Reviewer #2: Yes

Reviewer #3: Partly

3. Has the statistical analysis been performed appropriately and rigorously? 

Reviewer #1: Yes

Reviewer #2: N/A

Reviewer #3: I Don't Know

4. Have the authors made all data underlying the findings in their manuscript fully available?

Reviewer #1: Yes

Reviewer #2: Yes

Reviewer #3: No

5. Is the manuscript presented in an intelligible fashion and written in standard English?

Reviewer #1: Yes

Reviewer #2: Yes

Reviewer #3: Yes

6. Review Comments to the Author

Reviewer #1: The author has seamlessly incorporated all the comments. They carefully considered each suggestion and integrated it seamlessly into their revised manuscript.

Reviewer #2: Authors revised the manuscript according to reviewers comments and suggestions. Manuscript quality is enhanced and its scientific value increased. However, going through the manuscript one can easily spot few lingual and grammatical issues which needs to be addressed. so, authors suggested to get the manuscript proofread by an English native speaker to avoid such issues. Also, authors needs to double check the references style which has to meet with the journal one.

Reviewer #3: Authors revised the manuscript, mostly, according to reviewer comments. My comments for the manuscript:

1. CM-SPAM algorithm is used to first find the frequent sequential pattern in data that are then visualized with proposed three methods (NBVFS-WG, NBVFS-CM and NBVFS-TM). The topic is interesting and the problem is clearly stated. Authors claim that the “proposed visualization methods can significantly lower the time of information processing, for example, during the creation of machine learning models”. I did not find discussion about this claim in the results section. Similarly, lines 59-60: “without eliminating any relevant information during post-processing”. This line needs some more detail.

2. In abstract: Multiply or multiple?

3. Why CM-SPAM is used as there are other SPM algorithms too? In CM-SPAM, only minsup is the must parameter, other parameter such as min no of patterns, max number of patterns and gap are optional parameters. Authors should discuss why they used optional parameters. These are used to reduce he total number of discovered patterns? It is interesting that max gap of 2 is used. This will enforce the CM-SPAM to find patterns with some gaps and the order of events in the discovered patterns will change. I believe in the two datasets the order is important so why max gap of 2 is used? By the way add the reference for CM-SPAM.

Fournier-Viger, P., Gomariz, A., Campos, M., Thomas, R. (2014). Fast Vertical Mining of Sequential Patterns Using Co-occurrence Information. Proc. 18th Pacific-Asia Conference on Knowledge Discovery and Data Mining (PAKDD 2014), Part 1, Springer, LNAI, 8443. pp. 40-52.

4. Figures quality is very low. Should be replaced with high quality images.

5. Add the stats for the two datasets in a table form for easy understanding. Also add their references.

6. The data is converted to integer form to apply SPM algorithms. Right? If so then also explain the transformation process. Figures are hard to understand as it is hard to know which integers stands for which event or item.

7. This line: For example: the order of 306 actions is always 2 → 3 → 4 → 8; if actions 2,3 and 4 occur, it is almost 100% that 8 307 will occur as well. This kind of rules we can get with sequential rules. How authors are sure that 100% time, 8 will follow 2,3, and 4?

8. Ref [2] and [3] seems old. Authors can replace them recent references for the application of SPM. For example:

to extract hypernym relations from texts (Aldine et al. A 3-phase approach based on sequential mining and dependency parsing for enhancing hypernym patterns performance. The Knowledge Engineering Review, 36: E13, 2021.)

for genome analysis (Nawaz et al. Using artificial intelligence techniques for COVID-19 genome analysis. Applied Intelligence, 51: 3086-3103, 2021)

for tourist movement analysis (Cheng et al. A sequential pattern mining approach to tourist movement: The case of a mega event. Journal of Travel Research, 62(6): 1237-1256, 2023)

for workload prediction in cloud environment (M Amiri, L. Mohammad-Khanli and R. Mirandola. A sequential pattern mining model for application workload prediction in cloud environment. Journal of Network and Computer Applications, 105:21-62, 2018)

for mining discrete clinical data (Estiri, S. Vasey and S. N. Murphy. Transitive Sequential Pattern Mining for Discrete Clinical Data. In Proceedings of AIME, pp. 414-424, 2020)

for malware behavior analysis (Nawaz et al. MalSPM: Metamorphic malware behavior analysis and classification using sequential pattern mining. Computers & Security, 118: 102741, 2022)

9. It would be better if authors provide the GitHub link for the developed code

7. PLOS authors have the option to publish the peer review history of their article (what does this mean?). If published, this will include your full peer review and any attached files.

Reviewer #1: **Yes: **Shashank Singh

Reviewer #2: No

Reviewer #3: No

---

## [Author Response · Author response to Decision Letter 1]

30 Jan 2024

Dear Reviewers,

thank you for your valuable comments and suggestions. You can find our response letter attached to the submission.

Best regards

László Bántay

---

## [Decision Letter · Decision Letter 2]

23 Feb 2024

PONE-D-23-17033R2Network-based visualisation of frequent sequencesPLOS ONE

Dear Dr. Bántay,

Thank you for submitting your manuscript to PLOS ONE. After careful consideration, we feel that it has merit but does not fully meet PLOS ONE’s publication criteria as it currently stands. Therefore, we invite you to submit a revised version of the manuscript that addresses the points raised during the review process.

We look forward to receiving your revised manuscript.

Kind regards,

Praveen Kumar Donta, Ph.D.

Academic Editor

PLOS ONE

Journal Requirements:

Reviewers' comments:

Reviewer's Responses to Questions

**Comments to the Author**

1. If the authors have adequately addressed your comments raised in a previous round of review and you feel that this manuscript is now acceptable for publication, you may indicate that here to bypass the “Comments to the Author” section, enter your conflict of interest statement in the “Confidential to Editor” section, and submit your "Accept" recommendation.

Reviewer #2: All comments have been addressed

Reviewer #3: (No Response)

Reviewer #4: All comments have been addressed

2. Is the manuscript technically sound, and do the data support the conclusions?

Reviewer #2: Yes

Reviewer #3: Yes

Reviewer #4: Yes

3. Has the statistical analysis been performed appropriately and rigorously? 

Reviewer #2: N/A

Reviewer #3: I Don't Know

Reviewer #4: Yes

4. Have the authors made all data underlying the findings in their manuscript fully available?

Reviewer #2: Yes

Reviewer #3: Yes

Reviewer #4: No

5. Is the manuscript presented in an intelligible fashion and written in standard English?

Reviewer #2: Yes

Reviewer #3: Yes

Reviewer #4: Yes

6. Review Comments to the Author

Reviewer #2: The authors have appropriately addressed and corrected all the issues as per my previous comments. The related work has been enriched, and the indistinct description, as well as deficient analysis, have been improved and refined. More discussions have also been added. This paper has been comprehensively improved in terms of correctness, completeness, and readability to reach the standard for publication.

Reviewer #3: Authors properly addressed most of my previous concerns/suggestions.

Following previous comments needs revision from authors.

Comment 1. Authors did not provide justification for their claim that how the poposed

apptoach significantly reduce the computational time in information processing

Comment 3: Authors should provide the reason for why CM-SPAM is used and not other algorithms

such as CM-SPADE or TKS?

Comment 7: The pattern 2 3 4 and 8 occurs for which CM-SPAM generates the SUP.

100% occurences of such patterns means that support is equal to total number of records.

If it is the case then author can keep the line. Otherwise author can rewrite the line to remove

the 100% with SUP value.

Reviewer #4: Authors addressed all the comments and concerned carefully. The manuscript stands for the acceptance.

7. PLOS authors have the option to publish the peer review history of their article (what does this mean?). If published, this will include your full peer review and any attached files.

Reviewer #2: No

Reviewer #3: No

Reviewer #4: No

---

## [Author Response · Author response to Decision Letter 2]

12 Mar 2024

Dear Reviewers,

thank you for your additional comments and suggestions, we have done our best to amend the manuscript according to them. Please find the detailed answers to your findings below.

Comment 1. Authors did not provide justification for their claim that how the proposed approach significantly reduce the computational time in information processing

Thank you for pointing on this topic. Our intention was not to claim that the method saves computational time, but it lowers the information processing time of a human. The related parts in the Abstract and in the Conclusion were changed according to this aspect.

Comment 3: Authors should provide the reason for why CM-SPAM is used and not other algorithms such as CM-SPADE or TKS?

Thank you for this note, the reason of the selection is discussed now on page 8/9. CM-SPADE gives similar performance, TKS does not have the minsup parameter.

Comment 7: The pattern 2 3 4 and 8 occurs for which CM-SPAM generates the SUP. 100% occurrences of such patterns means that support is equal to total number of records. If it is the case then author can keep the line. Otherwise author can rewrite the line to remove the 100% with SUP value.

Thank you for highlighting this potentially confusing part. The mentioned sentence on page 10 has been changed.

---

## [Editor Report · Decision Letter 3]

14 Mar 2024

Network-based visualisation of frequent sequences

PONE-D-23-17033R3

Dear Dr. Bántay,

We’re pleased to inform you that your manuscript has been judged scientifically suitable for publication and will be formally accepted for publication once it meets all outstanding technical requirements.

Kind regards,

Praveen Kumar Donta, Ph.D.

Academic Editor

PLOS ONE
---

## [Editor Report · Acceptance letter]

26 Apr 2024

PONE-D-23-17033R3 

PLOS ONE

Dear Dr. Bántay, 

I'm pleased to inform you that your manuscript has been deemed suitable for publication in PLOS ONE. Congratulations! Your manuscript is now being handed over to our production team.

Kind regards, 

on behalf of

Dr. Praveen Kumar Donta 

Academic Editor

PLOS ONE